# Successfully Managed Respiratory Insufficiency in a Patient with a Novel Pathogenic Variant of the *BMPER* Gene: A Case Report

**DOI:** 10.3390/diagnostics12030626

**Published:** 2022-03-03

**Authors:** Ho Eun Park, Jin A. Yoon, Yong Beom Shin

**Affiliations:** Department of Rehabilitation Medicine, Pusan National University School of Medicine and Biomedical Research Institute, Pusan National University Hospital, Busan 49241, Korea; phe8948@gmail.com (H.E.P.); jinaa9606@gmail.com (J.A.Y.)

**Keywords:** bone morphogenetic protein-binding endothelial cell precursor-derived regulator (BMPER), diaphanospondylodysostosis (DSD), ischiospinal dysostosis (ISD), respiratory failure

## Abstract

Bone morphogenetic protein-binding endothelial cell precursor-derived regulator (*BMPER*) gene mutation presents a disease spectrum ranging from a mild type of ischiospinal dysostosis (ISD) to a more severe type of diaphanospondylodysostosis (DSD). It is known that *BMPER* gene mutations are very rare, and their resulting clinical manifestations, including musculoskeletal modifications, appear in a spectrum of various types and severity levels. With the development of genetic diagnosis, case reports of patients with specific mutations in the *BMPER* gene have been published. The most commonly known clinical features are kidney structural problems, including neuroblastoma and renal cysts. Meanwhile, respiratory failure is a common and fatal symptom for patients with *BMPER* gene mutation, but it does not appear to have been well evaluated or managed so far. We report a case of a confirmed novel mutation of c.1750delT (p.Cys584fs) in the *BMPER* gene in a female adolescent patient and highlight the importance of the regular assessment of respiratory failure for successful management of this condition.

## 1. Introduction

Diaphanospondylodysostosis (DSD, cytogenetic location 7p14.3, phenotype MIM number 608022) and ischiospinal dysostosis (ISD) are extremely rare and fatal diseases caused by biallelic mutations in the bone morphogenetic protein (*BMP*)-binding endothelial cell precursor-derived regulator (*BMPER*) gene [1,2,3]. BMPs are now known to play important roles in the formation and maturation of organs derived from mesenchymal stem cells, such as bone, cartilage, muscle, kidney, blood vessels, and other tissues [4]. BMPER has been known to function as both enhancer and antagonist of *BMP* signals and endothelial cell differentiation, which have a critical role in maturation of the vertebrae, kidneys, and other tissues [5,6]. Patients with DSD or ISD have diverse clinical features, such as abnormal ossification and segmentation of vertebrae, missing ribs, short neck and stature, bell-shaped thorax, compromised craniofacial features, possible nephrogenic cysts or tumors, and respiratory insufficiency.

DSD and ISD were classified as separate diseases in the past, but recently they have been considered as a single disease spectrum with various clinical manifestations. Patients with typical DSD, the fatal phenotype, present with severe respiratory failure during the perinatal period and inevitably undergo invasive ventilation. To date, only approximately 30 patients have been reported to have DSD and ISD [7,8,9].

However, for patients with attenuated DSD, respiratory insufficiency seems to be overlooked in comparison to other abnormal signs, such as skeletal dysplasia or renal abnormalities. As respiratory problems are not identified early in life, the need for further evaluation and management of respiratory problems has not been discussed.

In this case study, we report a novel pathogenic variant of the *BMPER* gene identified through Sanger sequencing in a patient showing a less severe phenotype of DSD that might have been previously classified as attenuated DSD. In addition, we share our experience of coping with respiratory failure in the patient without invasive procedures through regular evaluation of respiratory function.

Written parental informed consent, as well as written adolescent assent, was obtained.

## 2. Case Report

A 17-year-old girl, the first child born to a healthy non-consanguineous Korean couple after an uncomplicated pregnancy, was born full-term through elective cesarean section due to breech presentation and weighed 2600 g (<10%) at birth. Some abnormal morphologies, such as short neck and protruding abdomen, were remarkable, but no abnormal findings were notable in the chromosome study and the ultrasonography of the brain and the abdomen.

From infancy to pre-school age, she suffered from repetitive respiratory infections, atelectasis, and chest retraction. She was hospitalized several times and had two respective events of intubation, followed by barely successful extubations. During her childhood, her morphologic features were prominent, with facial dysmorphism of a short nose, depressed nasal bridge, limited neck movement, low-set hair line, short stature, bell-shaped thorax, and protruded abdomen. Computed tomography (CT) of the whole spine and plain radiography showed multiple skeletal anomalies (Figure 1).

At 12 years of age, she was admitted to the intensive care unit due to sudden respiratory failure and worsening of pneumonia, and she was referred to our department for pulmonary rehabilitation. Her first pulmonary function test showed a pattern of severe restrictive lung disease and low respiratory muscle strength and coughing power (Table 1). We trained her on the air stacking exercise using a bag valve mask for lung volume recruitment and assisted cough for sputum expectoration, although she did not complain of any hypoventilation symptoms and had a normal arterial blood analysis (ABGA).

At the same time, we tried Sanger sequencing of the patient and her mother to determine the cause of multiple anomalies and respiratory failure. Approximately 66 million paired-end 76-base pair reads were generated. Paired-end read exome sequencing of the patient generated at least 10-fold coverage for 94.86% of the targeted regions. The novel pathogenic variant allele, c.1750delT (p.Cys584fs), was identified, which resulted in a biallelic *BMPER* gene mutation (Figure 2 and Table 2). Subsequent Sanger sequencing of her mother showed a nonsense variant of c.1672C>T (p.Arg558*), previously known to be the causative mutation allele. Her father’s information was unavailable.

**Table 2 diagnostics-12-00626-t002:** Summary of clinical manifestations and genetic mutations of previously reported patients with ISD and DSD and the patient in the current study.

Patient Number	Gender	Survival	Consanguinity	Ethnicity	Respiratory Distress at Birth	Kidney Pathology	Other Abnormalities	Axial Skeletal Anomalies	cDNA	Protein	Clinical Diagnosis	Reported Year [Reference Number]	
1	Female	>33 years	No	Japanese	None	NB	Developmental delay, Paraparesis, feet deformities	Vertebral segmentation defects, rib anomalies, ischial hypoplasia, sacral hypoplasia	NA	NA	ISD	1999 [10]	
2	Male	>7 years	No	Japanese	None	NoM	Developmental delay, Facial dysmorphism, Paraparesis, feet deformities	Vertebral segmentation defects, ischial hypoplasia, sacral hypoplasia	NA	NA	ISD	1999 [10]	
3	Female	>38 years	1st degree	Japanese	None	NoM	Paraparesis, feet deformities	Vertebral segmentation defects, rib anomalies, ischial hypoplasia, sacral hypoplasia	NA	NA	ISD	1999 [10]	
4	Male	>3 months	No	Japanese	Yes, intubation	NoM	Coxa valga	Vertebral segmentation defects, rib anomalies, ischial hypoplasia, sacral hypoplasia	NA	NA	ISD	1999 [10]	
5	Male	>6 months	No	Japanese	None	NoM	Ichthyosis, cryptorchidism, inguinal hernia, heart anomaly, alopecia	Vertebral segmentation defects, rib anomalies, ischial hypoplasia	NA	NA	ISD	1999 [10]	
6	Male	>10 months	No	Caucasian	None	Polycystic, NB	Facial dysmorphism, macrocephaly	Vertebral segmentation defects, rib anomalies, ischial hypoplasia	NA	NA	ISD	2001 [11]	
7	Male	>2 years	No	Korean	None	Polycystic	Developmental delay, dislocated knee, and hip joints	Vertebral segmentation defects, rib anomalies, ischial hypoplasia, sacral hypoplasia	NA	NA	ISD	2003 [12]	
8	Female	>5 years	No	Japanese	None	Monocystic	Cleft palate, Developmental delay, brachymesophalangia	Vertebral segmentation defects, rib anomalies, ischial hypoplasia	NA	NA	ISD	2003 [12]	
9	Male	Stillborn	Yes	Mali	-	Polycystic, NB	Feet deformities, Nail hypoplasia	Vertebral segmentation defects, rib anomalies, ischial hypoplasia	NA	NA	DSD	2005 [1]	
10	Female	Stillborn	Yes	Mali	-	Polycystic, NB	Genu recurvatum	Vertebral agenesis	NA	NA	DSD	2005 [1]	
11	Male	Neonatal death	No	NoM	Yes, fatal	NoM	Facial dysmorphism, Nail hypoplasia	Vertebral ossification defects	NA	NA	DSD	2005 [1]	
12	Male	Neonatal death or stillborn	No	NoM	NoM	NoM	Cleft palate	Vertebral ossification defects	NA	NA	DSD	2005 [1]	
13	Male	Neonatal death	No	Hispanic	Yes, fatal	Polycystic, NB	Facial dysmorphism	Vertebral ossification and segmentation defects, rib anomalies, ischial hypoplasia, sacral agenesis	NA	NA	DSD	2007 [13]	
14	Male	Neonatal death	No	European	Yes, fatal	NoM	Facial dysmorphism, Inguinal hernia	Vertebral ossification defects, sacral agenesis	NA	NA	DSD	2007 [13]	
15	Female	Neonatal death	NoM	Caucasian	Yes, intubation	Polycystic	Tracheomalacia	Vertebral ossification defects, rib anomalies, sacral agenesis	NA	NA	DSD	2007 [13]	
16	Male	5 years	No	European	Yes, intubation	Polycystic, Wilms tumor	Facial dysmorphism, Hearing loss, macrocephaly, Peripheral neuropathy due to spinal cord anomaly	Vertebral ossification defects, rib anomalies, sacral agenesis	c.26_35del10ins14 c.1032+5G>A	p.Ala9Glufs*4	DSD	2007 [13], 2010 [2], 2012 [14,15]	
17	Female	15 months	2nd degree	Arabic	Yes, intubation	Polycystic	Facial dysmorphism	Vertebral ossification and segmentation defects, rib anomalies, sacral agenesis	c.310C>T	p.Gln104*	DSD	2011 [3]	
18	Female	4 months	No	Arabic	Yes, NICU care	Normal	Nail hypoplasia	Vertebral ossification and segmentation defects, rib anomalies, sacral agenesis	c.310C>T	p.Gln104*	DSD	2011 [3]	
19	Male	>13 years	No	British-European	None	Normal	Cleft palate, Facial dysmorphism	Scoliosis, Vertebral ossification defects, rib anomalies	c.251G>T c.1078+5G>A	p.Cys84Phe	Attenuated DSD	2015 [9]	
20	Male	>6~13 years	No	British-European	None	Normal	Cleft palate, Facial dysmorphism	Scoliosis, Vertebral ossification defects, rib anomalies	c.251G>T c.1078+5G>A	p.Cys84Phe	Attenuated DSD	2015 [9]	
21	Male	>6 years	No	British-European	None	Normal	Cleft palate, Facial dysmorphism	Scoliosis, Vertebral ossification defects, rib anomalies	c.251G>T c.1078+5G>A	p.Cys84Phe	Attenuated DSD	2015 [9]	
22	Female	>2 years	No	Swedish	None	Normal	Facial dysmorphism, Hearing loss	Vertebral segmentation defects, rib anomalies, ischial hypoplasia, sacral hypoplasia	c.416C>G c.942G>A	p.Thr139Arg p.Trp314*	ISD	2016 [16]	
23	Male	>19 years	No	Korean	Yes, oxygen	Hydronephrosis	Facial dysmorphism, Paraparesis, feet deformities, neurogenic bladder, flat acetabulum, coxa valga	Vertebral segmentation defects, rib anomalies, ischial hypoplasia	c.1672C>T c.1988G>A	p.Arg558* p.Cys663Tyr	ISD	2016 [16]	
24	Male	>9 years	No	NoM	Yes	Renal caliectasis	Facial dysmorphism, Muscle wasting, hypertension, hypercalcinuria, coxa valga	Vertebral segmentation defects, rib anomalies, ischial hypoplasia, sacral hypoplasia	c.322T>C	p.Cys108Arg/7p14.3p14.2del	DSD	2017 [7]	
25	Male	Fetus (terminate)	No	NoM	-	Nephroblastomatosis	Facial dysmorphism	Vertebral ossification defects, rib anomalies	c.496T>A c.501_502delGT	p.Cys166Ser p.Phe168*	DSD	2018 [17]	
26	Male	>2 years	3rd degree	NoM	NoM	Renal calculus, hydronephrosis, congenital megaureter	Microcephaly, developmental delay, flat acetabulum, coxa valga	Vertebral segmentation defects, rib anomalies, ischial hypoplasia, sacral hypoplasia	c.314G>A	p.Cys105Tyr	ISD	2019 [8]	
27	Male	Fetus (terminate)	No	Jewish	-	Cystic	Skull malformation	Spine, chest malformation	NA	NA	DSD (pressumed)	2019 [18]	
28	Male	Fetus (terminate)	No	Jewish	-	Normal	Feet deformities	Spine malformation	NA	NA	DSD (pressumed)	2019 [18]	
29	Female	Fetus (terminate)	No	Jewish	-	Normal	Short trunk, distended abdomen, skull bone ossification defects	Vertebral ossification defects	c.410T>A	p.Val137Asp	DSD	2019 [18]	
30	Female	>17 years	No	Korean	None	Polycystic	Facial dysmorphism	Vertebral segmentation defects, rib anomalies, ischial hypoplasia, sacral hypoplasia	c.1672C>T c.1750delT	p.Arg558* p.Cys584fs	Attenuated DSD	Current study	

Inequality signs indicate that the patient survived longer than the age mentioned. Age without an inequality sign indicates the age at which the patient died, followed by the cause of death. ISD, ischiospinal dysostosis; DSD, diaphanospondylodysostosis; RSV, respiratory syncytial virus infection; BL, birth length; NoM, not mentioned; NB, nephroblastomatosis; NA, not available, *, termination codon; fs, type of change is a frameshift.

Annual pulmonary function tests and sleep studies were performed to determine the exact timing of deterioration of her respiration. At 14 years of age, a sharp deterioration in respiration was remarkable, as the forced vital capacity (FVC) reached less than 20% of the reference value. Although she still had no complaints, we trained her to apply a non-invasive ventilator (NIV) during the night to prevent hypoventilation.

At the age of 15 years, she had an upper respiratory infection and visited the emergency room with dyspnea and drowsiness. ABGA showed marked carbon dioxide retention of up to 73 mmHg in arterial partial pressure of carbon dioxide (pCO_2_), 40 mmHg in arterial partial pressure of oxygen (pO_2_), and 73% of arterial oxygen saturation (SaO_2_), indicating definite hypoventilation. Under daily serial percutaneous CO_2_ monitoring and ABGA, we struggled to help her sputum expectoration and lung recruitment using the mechanical insufflation–exsufflation device and NIV. She was able to avoid intubation and invasive ventilation, and her breathing improved within a week of hospitalization. Finally, she recovered consciousness and was able to wean off daytime ventilatory support.

## 3. Discussion

DSD and ISD are rare, inherited autosomal recessive diseases with biallelic pathogenic variants of the *BMPER* gene [3]. BMPER regulates BMP signaling and the coupling of osteogenic differentiation and vascularization of mesenchymal stem cells, ultimately affecting bone and organ formation [4,5,6]. Patients with ISD may present with segmentation and ossification defects of the axial skeleton, resulting in fused or butterfly-shaped vertebrae, ischiosacral hypoplasia, and absent or fused ribs. DSD is an extremely rare and life-threatening disorder, with a known incidence of less than 1 per 1,000,000 pregnancies [17]. Patients with DSD have more severe skeletal malformations than those with ISD, often leading to intrauterine or neonatal death due to respiratory insufficiency. In addition, there may be accompanying critical kidney problems, so ultrasonographic examination is required.

Although DSD and ISD were regarded as different diseases in the past, the recent perspective is that it is reasonable to consider these two diseases as a single disease spectrum of varying severity with overlapping clinical features [7,8,9]. This case report supports the previously proposed hypothesis that DSD, attenuated DSD, and ISD should be considered as a disease spectrum caused by *BMPER* gene mutations. Clinical diagnosis by case was classified into three types for convenience in Table 2, however, we would like to emphasize that the extension of phenotype indicates a spectrum disorder and is not intended to classify it into subgroups such as DSD, attenuated DSD, and ISD.

This case is significant because it reports a new pathogenic variant due to a frameshift mutation in the *BMPER* gene, causing a rare disease. Sanger sequencing confirmed a novel pathogenic variant of p.Cys584fs caused by a frameshift mutation, and this induced heterozygous biallelic variations in the *BMPER* gene, along with the p.Arg559Ter nonsense mutation. As this is an autosomal recessive inherited disease, it is presumed that this novel pathogenic variant was inherited from the patient’s father. However, a limitation of this study is that the father could not be tested due to his unavailability.

As the number of reported patients is still small, about 30, the factors determining the severity of the disease have not been clearly identified. It was assumed that the type of mutation could affect the severity of the disease by elucidating the mutation type of siblings, considering the characteristics of a disease with an autosomal recessive inheritance pattern due to biallelic mutation [8]. In addition, patient number 23 shared the same allele mutation c.1672C>T as the patient reported in the current study, but had a different disease severity. Therefore, we suggest that c.1988G>A in patient 23 with ISD phenotype is a less lethal mutation, and c.1750delT, a frameshift mutation in the patient in the current study with attenuated DSD, is a more lethal one.

However, it is difficult to affirm that the type of mutation itself determines the severity of the disease. Although there is a limitation in that genetic testing was performed on only one patient, patient 28, it is highly likely that patients 26 and 27, who are in a sibling relationship, also had the same mutation. Nevertheless, the three patients showed many differences in clinical features, such as kidney pathology and skeletal abnormalities. Therefore, it is expected that there will be unknown modifiers that affect disease severity, in addition to the mutation types. Several studies have indicated that typical DSD is fatal, especially in the perinatal period, and may be accompanied by renal abnormalities, such as renal cysts, hydronephrosis, and nephroblastoma, also known as Wilm’s tumor [1,2,13,14,15]. However, there are only a few reports on the evaluation and management of respiratory failure in the form of restrictive lung disease occurring in the long-term course of the disease. The patient, in this case, had poor pulmonary function due to a thoracic deformity, which resulted in hypoventilation.

To the best of our knowledge, this is the first report highlighting the importance of regular assessment and timely intervention for respiratory failure in patients with *BMPER* gene mutations. We predicted that the patient may develop respiratory failure in the pattern of restrictive lung disease due to chest wall deformity and scoliosis, so we monitored changes in her basal respiratory function annually. Even though respiratory failure progressed as upper respiratory infection occurred, she avoided intubation, tracheostomy, and invasive mechanical ventilation through appropriate management based on objective information.

In the normal population, FVC shows an upward trend with age, peaks around the age of 20, and then decreases [19]. There have been no previous studies on pulmonary function values in reference to age in patients with restrictive lung diseases, such as neuromuscular disease or chest wall disorder. This may be because the degree of disease progression and thoracic cage deformity affects respiratory function significantly more than age. The exact timing of pulmonary rehabilitation interventions, including ventilator support in these patients, is not clearly defined, but nocturnal mechanical ventilation is known to reduce long-term mortality and unplanned hospitalization.

According to the DMD treatment guidelines, lung volume recruitment is recommended if the FVC falls below 60% of the predicted value. Assisted coughing is recommended if the FVC is less than 50% of the predicted value and the PCF is less than 270 L/min, or if the MEP is less than 60 cmH_2_O. Nocturnal assisted ventilation is required when there are signs or symptoms of hypoventilation, sleep-related respiratory abnormalities, FVC < 50% of predicted value and MIP of 60 cmH_2_O, or awake baseline SpO_2_ > 95% or pCO_2_ > 45 mmHg. Assisted daytime ventilation should be added if daytime hypoventilation is confirmed despite nocturnal ventilation [20,21].

Although we did not strictly comply with these guidelines, we tried to manage her respiratory failure based on these guidelines, considering her compliance and subjective symptoms. As a result, even though her respiratory function worsened rapidly, following URI at the age of 15, invasive ventilation could be avoided. This was possible because we were aware of her baseline respiratory function before her condition worsened, and adequate respiratory rehabilitation treatment was provided promptly. Fortunately, she still maintains non-invasive ventilation only at night and performs most of her daily activities independently.

This case is the first in which respiratory failure was managed based on regular assessment and management of respiratory functions, even in acute exacerbation in a patient with *BMPER* gene mutation. We suggest that respiratory insufficiency due to skeletal deformities also affects the disease course with a restrictive lung disease pattern. Therefore, we recommend early pulmonary rehabilitation to maintain chest compliance and prevent the acute deterioration of respiratory function. It is expected that pulmonary rehabilitation, such as lung volume recruitment, sputum toileting, and ventilatory support via NIV, will improve the quality of life of patients and reduce hospitalization and duration of hospital stay.

Future studies are needed to follow up on long-term progression throughout life and determine the role of pulmonary rehabilitation as a disease-modifying therapy for patients with *BMPER* gene mutations. Moreover, as more patients are reported in the future, it is necessary to analyze the factors that determine the severity of the disease.

## Figures and Tables

**Figure 1 diagnostics-12-00626-f001:**
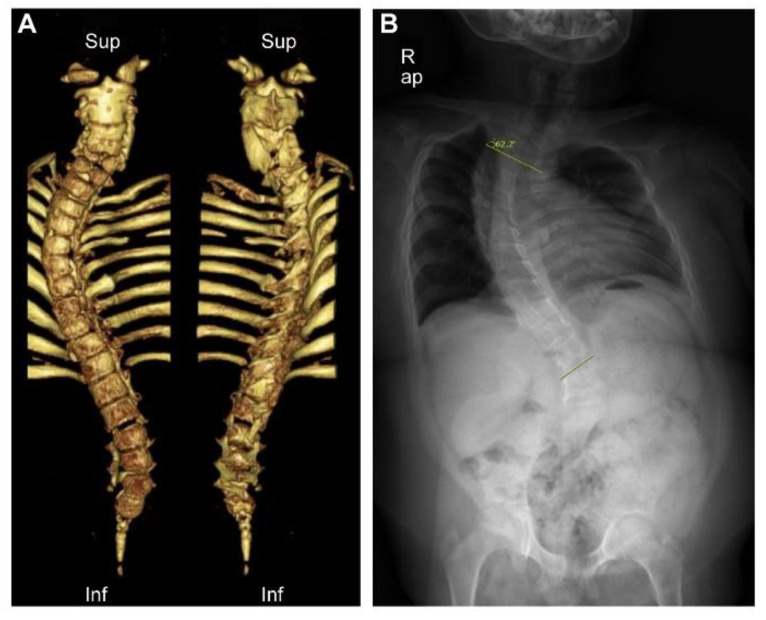
Three-dimensional computed tomography and plain radiography of the whole spine. (**A**) Multiple anomalies of the spine and the sacrum, including fusion of C2/3, C4/5 vertebra, butterfly vertebra of C5 to C7, dysraphism of C1, C5, C6, and T8, and hypogenesis of the sacrum, were notable. (**B**) Severe scoliosis of Cobb angle of 62°, decreased number of ribs, and downward tilt of ischiopubic rami were identified.

**Figure 2 diagnostics-12-00626-f002:**
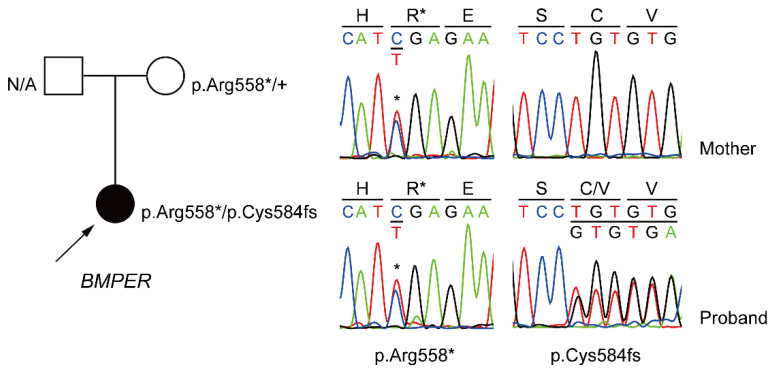
Family pedigree and mutation in the patient and Sanger sequencing traces confirming the compound heterozygous variants in the *BMPER* gene. *, termination codon; fs, type of change is a frameshift.

**Table 1 diagnostics-12-00626-t001:** Result of annual pulmonary function tests of the patient.

	Age of the Patient (Years)					
	12	13	14	15	16	17
FVC † (mL)	320	500	390	280	420	550
FVC (% reference)	28	25.02	18.68	13.39	20.09	25.98
FEV1 † (mL)	300	400	330	220	340	420
FEV1 (% reference)	30	22.57	18.13	12.12	19.05	23.30
FEV1/FVC	93.75	80	84.61	78.57	80.95	76.36
PCF † (L/min)	60	70	60	60	90	70
MIP † (cmH_2_O)	45	24	42	30	40	40
MEP † (cmH_2_O)	40	36	36	60	63	65
Height (cm)	111	118	120	120	120	120
Weight (kg)	38	49.2	54	55	59.5	60
BMI (kg/cm^2^)	30.84	35.33	37.5	38.20	41.32	41.67

† FVC, forced vital capacity; FEV1, forced expiratory volume in 1 s; PCF, peak cough flow; MIP, maximal inspiratory pressure; MEP, maximal expiratory pressure; BMI, body mass index.

## Data Availability

The data that support the findings of this study are available on request from the corresponding author. The data are not publicly available due to privacy or ethical restrictions.

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
