# Peer review of "Successfully Managed Respiratory Insufficiency in a Patient with a Novel Pathogenic Variant of the BMPER Gene: A Case Report"

_diagnostics, 2022, doi:10.3390/diagnostics12030626_

Round 1

Reviewer 1 Report

Dear authors,

 Thank you for giving me an opportunity of reviewing this case report.This report described a rare case with diaphanospondylodysostosis with a novel mutation of BMPER gene. The authors offered the detailed respiratory function data and the clinical data in other reports of DSD patients. These data can recognize the disease severity and clinical course in attenuated type of DSD.

I would like to request the authors to describe some detailed information as the following.

1.Please describe the patient's stature data such as height and weight.

2.Please offer the respiratory setting including oxygen concentration and pressure.

3.Please discuss the speculation or the reason why the patients exhibited the attenuated type despite of a novel mutation of BMPER mutation leading to frameshilt.

Author Response

Response to Reviewer 1 Comments

Point 1: Please describe the patient's stature data such as height and weight.

Response 1: We supplemented the body measurement data by adding height, weight, and body mass index (BMI) to Table 1, and revised the table title. (Please see Table 1 and line 83)

Point 2: Please offer the respiratory setting including oxygen concentration and pressure.

Response 2: We presented oxygen partial pressure and oxygen concentration values from arterial blood gas analysis (ABGA) at the time she developed severe hypoventilation. (Please see lines 109-110)

Point 3: Please discuss the speculation or the reason why the patients exhibited the attenuated type despite of a novel mutation of BMPER mutation leading to frameshilt.

Response 3: In addition to reporting a novel pathogenic variant, we had many concerns about why this patient could be classified as an attenuated type and what factors determined the severity of the disease. The reason that we did not discuss such content enough in the discussion was because we were afraid that our intention to emphasize the concept of the spectrum disorder and the importance of appropriate treatment for respiratory failure in this patient might be overshadowed. However, as you pointed out, since there have been few reports yet, we believe that sufficient discussion is necessary on the factors that determine the severity of the disease, so we expressed our opinion in the discussion. (Please see lines 138-143, 154-169, 227-228)

Reviewer 2 Report

The manuscript submitted to Diagnostics entitled “Successfully Managed Respiratory Insufficiency in a Patient 2 with a Novel Pathogenic Variant of the BMPER Gene: A Case Report” is a case study article in which authors are suggesting management of respiratory failure in a patient with BMPER gene mutation. As a reviewer, I find this article interesting but major revisions are needed.

First of all, there is an insufficient number of literature references. In the manuscript, many sentences are without literature references. For example: „Several studies have indicated….“ – all studies should be listed in the revised manuscript. Also, the introduction should be written more thoroughly, explaining the biology of malfunctioning BMPER protein and how inhibition of binding BMP molecules affect the pathology of the disease. I would also suggest describing the methodology more thoroughly.

Second of all, Table 2 is not very comprehensible. From where are all these patients from Table 2? Are those patients collected from literature? If so, for every patient should be a literature reference. Table with cases should be elaborated more in the discussion. 

Author Response

Response to Reviewer 2 Comments

Point 1: First of all, there is an insufficient number of literature references. In the manuscript, many sentences are without literature references. For example: „Several studies have indicated….“ – all studies should be listed in the revised manuscript. Also, the introduction should be written more thoroughly, explaining the biology of malfunctioning BMPER protein and how inhibition of binding BMP molecules affect the pathology of the disease. I would also suggest describing the methodology more thoroughly.

Response 1:

We have presented all references and have annotated the relevant content of the text. (Please see the main text and references)

We have elucidated the role of the BMP and BMPER protein and described the molecular and pathological mechanisms that affected this disease in the part of introduction. (Please see the first paragraph of introduction and discussion)

We have added contents to reinforce the methodology related to Sanger sequencing. (Please see lines 86-88)

Point 2: Second of all, Table 2 is not very comprehensible. From where are all these patients from Table 2? Are those patients collected from literature? If so, for every patient should be a literature reference. Table with cases should be elaborated more in the discussion.

Response 2: We supplemented Table 2 by presenting references that correspond to each patient, and have made more elaborated discussion about some of these cases. (Please see Table 2, lines 154-169 and references)

Round 2

Reviewer 2 Report

The manuscript submitted to Diagnostics entitled “Successfully Managed Respiratory Insufficiency in a Patient 2 with a Novel Pathogenic Variant of the BMPER Gene: A Case Report” is now suitable for publication, after the changes have been made.